# Activation of STAT3 and STAT5 Signaling in Epithelial Ovarian Cancer Progression: Mechanism and Therapeutic Opportunity

**DOI:** 10.3390/cancers12010024

**Published:** 2019-12-19

**Authors:** Chin-Jui Wu, Vignesh Sundararajan, Bor-Ching Sheu, Ruby Yun-Ju Huang, Lin-Hung Wei

**Affiliations:** 1Department of Obstetrics & Gynecology, National Taiwan University Hospital, College of Medicine, National Taiwan University, Taipei 10002, Taiwan; cjwu00@gmail.com (C.-J.W.); bcsheu@ntu.edu.tw (B.-C.S.); 2Cancer Science Institute of Singapore, National University of Singapore, Center for Translational Medicine, Singapore 117599, Singapore; csivsun@nus.edu.sg; 3Department of Obstetrics and Gynaecology, National University of Singapore, Singapore 119077, Singapore; rubyhuang@ntu.edu.tw; 4School of Medicine, College of Medicine, National Taiwan University, Taipei 10051, Taiwan

**Keywords:** ovarian cancer, STAT3, STAT5

## Abstract

Epithelial ovarian cancer (EOC) is the most lethal of all gynecologic malignancies. Despite advances in surgical and chemotherapeutic options, most patients with advanced EOC have a relapse within three years of diagnosis. Unfortunately, recurrent disease is generally not curable. Recent advances in maintenance therapy with anti-angiogenic agents or Poly ADP-ribose polymerase (PARP) inhibitors provided a substantial benefit concerning progression-free survival among certain women with advanced EOC. However, effective treatment options remain limited in most recurrent cases. Therefore, validated novel molecular therapeutic targets remain urgently needed in the management of EOC. Signal transducer and activator of transcription-3 (STAT3) and STAT5 are aberrantly activated through tyrosine phosphorylation in a wide variety of cancer types, including EOC. Extrinsic tumor microenvironmental factors in EOC, such as inflammatory cytokines, growth factors, hormones, and oxidative stress, can activate STAT3 and STAT5 through different mechanisms. Persistently activated STAT3 and, to some extent, STAT5 increase EOC tumor cell proliferation, survival, self-renewal, angiogenesis, metastasis, and chemoresistance while suppressing anti-tumor immunity. By doing so, the STAT3 and STAT5 activation in EOC controls properties of both tumor cells and their microenvironment, driving multiple distinct functions during EOC progression. Clinically, increasing evidence indicates that the activation of the STAT3/STAT5 pathway has significant correlation with reduced survival of recurrent EOC, suggesting the importance of STAT3/STAT5 as potential therapeutic targets for cancer therapy. This review summarizes the distinct role of STAT3 and STAT5 activities in the progression of EOC and discusses the emerging therapies specifically targeting STAT3 and STAT5 signaling in this disease setting.

## 1. Introduction

Epithelial ovarian cancer (EOC) is a heterogenous entity comprised of different histotypes [1] with unique molecular features and clinical characteristics that influence chemosensitivity and the probability of survival [2]. Cytoreductive surgery and platinum/taxane-combination chemotherapy remain the mainstay of primary treatment for advance-staged diseases, resulting in the initial remission in up to 80% of EOC patients. Nonetheless, approximately 75% of patients with advanced diseases develop recurrence within three years of diagnosis [3], which is generally not curable, owing to the development of chemoresistance. In this respect, EOC remains the most lethal of all gynecologic malignancies, and the relative survival rates at ten years for stage III and IV disease are 23% and 8%, respectively [4]. Molecular therapeutics targeting the angiogenesis and DNA damage repair pathways have provided significant steps forward in the management of certain EOC patients. However, there is still a clear unmet need for most patients with recurrence. Identifying novel molecular therapeutic targets relevant to the disease progression is thus highly anticipated.

Signal transducers and activators of transcription (STATs) belong to a family of cytoplasmic transcription factors that communicate signals from the cell membrane to the nucleus. The STAT family includes seven structurally and functionally related proteins: STAT1, STAT2, STAT3, STAT4, STAT5A, STAT5B, and STAT6 [5]. STATs have essential roles in fundamental processes, including sustaining proliferation, evading apoptosis, inducing angiogenesis, promoting invasion, and suppressing anti-tumor immunity [6,7]. Upon the binding of cytokines or growth factors to cognate receptors on the cell surface, STATs are tyrosine phosphorylated, particularly by Janus kinase (JAK), Abelson (Abl) kinase or SRC kinase families. Phosphorylated STAT (pY-STAT) dimer undergoes conformation change and shuttles into the nucleus, functioning as a transcription factor. Each STAT protein appears to have distinct physiologic functions in the development, differentiation, and immune response (For a comprehensive review, please see [7]). In particular, mice carrying homozygous deletion for *STAT5* (STAT5a^−/−^5b^−/−^) which later turned out to be hypermorphic *STAT5* deletion mice lacking the N-domains were infertile, with defects in the differentiation of functional corpora lutea, disrupting ovarian development [8].

STATs activation is rapid and transient under most physiological conditions. Notably, compelling evidence indicates that constitutive activation of STAT proteins, particularly STAT3 and STAT5, plays a critical role in oncogenic transformation. Clinically, aberrant activation of STAT3 and, to some extent, STAT5, is associated with both solid and hematopoietic cancers [9,10,11,12]. Accumulating evidence has indicated that downregulating STAT3/STAT5 mitigates the malignant behavior of cancer cells [13], highlighting the potential of STAT3/STAT5 as a therapeutic target. Collecting data has shown the role of STAT3 in the disease progression mechanism of EOC. Compared to normal or benign ovarian tumors, pY-STAT3/pY-STAT5 protein expression was significantly higher in the malignant EOC tissues, supporting its role in ovarian carcinogenesis [14,15]. The activation of the STAT3 pathway and the increase in pY-STAT3 (Tyr705) expression directly correlated with higher clinical stage, lower degree of differentiation, presence of lymph node metastasis, and more reduced survival in EOC [15,16,17]. Moreover, elevated pY-STAT3 expression in the omentum was associated with poor survival in patients with high-grade EOC. The activation and translocation of pY-STAT3 to the nucleus was observed in 29–58% of all EOC histotypes [13,16]. Specifically, nuclear pY-STAT3 expression was found to be associated with clear cell and serous carcinoma [17]. The activation of STAT3 pathway was, in particular, related to overall survival in ovarian clear cell carcinoma patients [16]. In recurrent diseases, levels of STAT3 activation were doubled, indicating that STAT3 activation could be directly associated with disease relapse [18]. Moreover, one study suggests STAT5 may be related to RELA (p65 subunit of NF-kB) and carboplatin resistance in EOC [19].

## 2. Regulation of STAT3/STAT5 Activation in EOC

Constitutive activation of STAT3/STAT5 has been identified in a wide range of human cancers. As a primary event during malignant transformation, somatic *STAT3* and *STAT5* driver mutations have been identified in hematopoietic neoplasms. For example, somatic mutations in the *STAT3* gene were found in 40% of granular lymphocytic leukemia and T-cell lymphoma patients, with recurrent mutations located on the gene segment encoding the SH2 domain, which mediates STAT3 dimerization and activation [20,21]. Also, a small percentage of granular lymphocytic leukemia patients harbored *STAT5B* mutations, resulting in increased transcriptional activity and phosphorylation [22]. However, genetic mutations that result in hyperactivated *STAT3/STAT5* have not been reported in EOC [23]. In EOC, constitutive upregulation of *STATs* in the absence of somatic mutations is primarily contributed through persistent Tyr phosphorylation signals. In general, *STAT3/STAT5* are activated in response to the binding of numerous cytokines, hormones, and growth factors to their receptors and by the activation of intracellular kinases, mostly in case of tyrosine phosphorylation by the four JAK family kinases. Typically, STAT3/STAT5 are activated by phosphorylation on critical residues (STAT3 Tyr residue 705 and Ser727 (ERK, JNK, and other stress kinases); STAT5A Tyr residue 694, Ser725 (CDK8) and Ser779 (PAK1/2) and STAT5B Tyr residue 699 and Ser730 (CDK8)) [9]. The JAK-STAT signaling in EOC can be further modulated by various molecular pathways, as summarized in Figure 1.

### 2.1. IL-6 Pathway

Upstream to the JAK-STAT signaling, the IL-6 family of cytokines is critical for signal transduction. The binding of IL-6 family cytokines to the ligand-binding subunit of gp130 initiates its homodimerization, activates JAK to bind to gp130, and then triggers downstream signaling cascades [24]. In EOC, The IL-6 family of cytokines is one of the major families of immunoregulatory cytokines. IL-6, LIF, and IL-11 secreted in the EOC tumor microenvironment function in concert to induce ovarian cancer cell JAK-STAT signaling [25,26,27].

### 2.2. TP53 Mutation

Notably, Tyr phosphorylation of JAK2 can be diminished by wild type but not mutant p53 in EOC cells [28]. Since there is a high frequency of somatic *TP53* mutation in high-grade serous carcinoma (HGSC), this suggests that STAT3 phosphorylation and its DNA binding activity can be modulated by the p53 status in HGSC.

### 2.3. Lipid Metabolism Pathway

STAT3 is also regulated by the cellular redox state controlling the transcription of genes related to the invasive phenotype. Pathways related to lipid metabolism known to affect the redox state thus are intriguing mechanisms since obesity is known to be a risk factor for poor EOC survival [29]. For example, leukotriene B_4,_ a lipoxygenase pathway metabolite, together with their cognate receptor leukotriene B_4_ receptor 2 (BLT2) contribute to the generation of reactive oxygen species (ROS) in EOC cells, which results in the activation of JAK and the STAT3-MMP2 cascade [30]. Leptin is an adipokine that exerts its activity through the membrane receptor, the obesity receptor (OB-R). The overexpression of OB-R in EOC significantly correlates with poor progression-free survival [31]. It has been shown that Leptin/OB-R signaling may phosphorylate STAT3 through the activation of JAK in EOC cells [32]. Recently, CD97, a member of the EGF-TM7 family of G-protein coupled receptors, is known to be expressed in several malignancies, including EOC. The interaction between overexpressed CD97 and its ligand CD55 activates JAK2/STAT3 signaling and confers an invasive cell phenotype of EOC [33].

### 2.4. Receptor Tyrosine Kinases (RTKs)

Several RTKs have been suggested to modulate the JAK-STAT signaling pathway. Vascular endothelial growth factor (VEGF), a key mediator of angiogenesis in EOC, induces pY-STAT3 through the binding of VEGF receptor 2 (VEGFR2) in EOC cells [34]. The expressions of VEGF, VEGFR1, and VEGFR2 are significantly correlated with pY-STAT3/pY-STAT5 in EOC [14]. Epidermal growth factor (EGF) is known either to directly activate JAK2/STAT3 signaling pathway or to induce the secondary mediator-like response mediated by the IL-6 axis in an autocrine manner in EOC [35]. EGFR which is overexpressed in EOC is significantly correlated with pY-STAT3 [17]. Oncogenic RAS and RAF mutations are prevalent in EOC, and these driver mutations are highly associated with aberrant ERK signaling, resulting in uncontrolled cellular proliferation [36,37]. Mechanistically, activated RTKs stimulate RAS activation, which then activates RAF. RAF phosphorylates and activated MEK, which in turn activates ERK through phosphorylation. Aberrant phosphorylation of ERK mediates phosphorylation of S727-STAT3 (Figure 1) and contributes to cisplatin resistance in certain EOC cell lines [38].

### 2.5. Alternative Cytokine or Non-Receptor Tyrosine Kinases

Granulocyte-colony stimulation factor (G-CSF), a commonly used cytokine receptor to aid hematopoietic recovery following chemotherapy, activates the JAK/STAT pathway in EOC through its cognate receptor, the G-CSFR, which is predominantly expressed in HGSC [39]. Furthermore, non-receptor tyrosine kinase SRC family members can alternatively tyrosine phosphorylate STAT proteins, and SRC is overexpressed and activated in late staged EOC. SRC family tyrosine kinases are essential for STAT activation in EOC, especially during metastasis. It is known that activated STAT localizes not only to nuclei, but also to focal adhesions. SRC family kinases induce pY-STAT3 and contribute to strong interactions between the pY-STAT3 and the focal adhesion complex [13]. In particular, c-SRC, a member of SRC family kinases, is primarily involved in hypoxia-triggered intracellular signaling. Under hypoxia, activation of c-SRC induces nuclear pY-STAT3 and enhances the binding ability of STAT3 to Hypoxia-inducible factor 1-alpha (HIF-1α), which contributes to chemoresistance in EOC [40]. Moreover, recombinant human erythropoietin (rhEpo) has been shown to bind to an alternative Epo receptor, EphB4, to activate the SRC-STAT pathway, triggering tumor growth, and resulting in decreased survival of EOC [41]. This further supports the notion that the use of rhEpo to treat anemia in cancer patients can compromise their overall survival [42].

### 2.6. Other Gene Regulatory Mechanisms

A dysregulated transcriptional control of STAT3 by miRNAs was reported in EOC. Increased expression of miR551b-3p, which is resultant from the frequent q26.2 amplification in HGSC, interacted with the *STAT3* promoter by recruiting RNA-pol-II and TWIST1 to turn on *STAT3* transcription [43]. The upregulation of STAT3 is subsequently required for miR551b-induced growth and metastasis of EOC cells. Enhancer of zeste homolog 2 (EZH2), a member of the polycomb repressor complex 2, functions primarily to promote transcriptional silencing via histone 3 on lysine 27 trimethylation (H3K27me3) and plays an essential role in EOC progression [44,45]. Interestingly, upon Tyr372 phosphorylation of EZH2 by protein kinase A, pY372-EZH2 efficiently interacted with STAT3 protein. This non-canonical EZH2 interaction reduced cellular levels of STAT3 and altered STAT3 activation, leading to the downregulation of downstream target IL-6R in EOC [46]. Furthermore, tyrosine phosphorylation of EZH2 by JAK3 in lymphoid neoplasia was reported to promote dissociation of the polycomb repressive complex 2 (PRC2) complex leading to decreased global H3K27me3 levels while it switches EZH2 to a transcriptional activator [47], but studies in EOC are lacking. Moreover, EZH2 can also interact with the STAT5 N-terminal oligomerisation domain, which was shown to be essential for B-cell acute leukemic transformation silencing the kappa light chain expression [48]. Such studies postulate that STAT3/5 interaction with EZH2 could be a valuable target interface for future therapeutic intervention [49], but future studies with EOC model systems are needed.

## 3. The Function of STAT3 and STAT5 in EOC

### 3.1. Apoptosis

Under normal physiological conditions, apoptosis is a process that governs programmed cell death and is responsible for the elimination of cells in normal tissues, to maintain tissue homeostasis. This process is commonly observed in several self-renewing tissues, including the gut, bone marrow, and skin, to accommodate newly generated cells daily. Emphasizing the crucial role of apoptosis in normal cell turnover, apoptosis remains to be one of the critical cell processes that are highly dysregulated during cancer progression. Evasion of apoptosis by cancer cells often results in excessive tumor growth, metastatic spread, and even causing resistance to cancer treatment. Apoptosis primarily occurs through two pathways. In the extrinsic pathway, binding of external death ligands to death receptors triggers the cascade, resulting in caspase-mediated cell death. While, in the intrinsic pathway, internal stimuli such as DNA damage, cellular stress triggers the activation of proapoptotic factors such as B-cell lymphoma 2 (BCL-2) family members that resemble three functional groups: inhibitors of apoptosis (BCL-2, BCL-_XL_, BCL-W, Mcl-1, BCL-B, and A1), promoters of apoptosis (BAX, BAK, and BOK); and regulatory proteins (BAD, BIK, BID, HrK, BIM, BMF, NOXA, and PUMA) [50]. Notably, tumor suppressor p53 is a transcription factor with primary pro-apoptotic function, and it remains one of the most commonly mutated genes in human cancers, underlining the significance of apoptosis deregulation in cancer [51].

Activated STAT3/5 proteins target specific inhibitors of apoptosis that mainly act on the intrinsic pathway to suspend cell death in cancer cells. Inhibition of STAT3 signaling leads to apoptosis of ovarian clear cell carcinoma and decreased BCL-2 expression [52]. G-CSF and Leptin can both activate STAT3 phosphorylation, and both can promote increased cellular BCL-2 levels, thereby protecting EOC cells against apoptosis [39,53]. RELA and STAT5 proteins transcriptionally activate the expression of *Bcl-x_L_* through direct promoter binding in ovarian, non-small-cell lung carcinoma and transformed leukemia cells, as well as render chemoresistance in carboplatin resistant ovarian cancer cell lines [19,54,55]. Also, cell survival signaling diminishes the effectiveness of chemotherapy, which contributes towards the acquisition and development of chemoresistance in cancer cells. In paclitaxel-resistant ovarian cancer cells, blocking of STAT3 activity suppresses STAT3 downstream antiapoptotic regulatory genes *BCL2L1*, *MCL1*, and *BIRC5*, which increases paclitaxel sensitivity in paclitaxel-resistant ovarian cancer cells in vitro [56]. Similarly, abrogation of constitutive STAT3 activity shows significant reductions in the expression of the BCL-2, BCL-_XL_, and Survivin protein, which circumvents cisplatin resistance in EOC [57].

MicroRNAs contribute to diverse physiological and pathological processes by involving in the regulation of several essential biological processes. An array of microRNAs induces apoptosis in cancer cells through JAK/STAT signaling in multiple cancer types. miR-17-5p, miR-133b, miR-134, miR-13, miR-147, miR-182, miR-204, miR-874 are among a few miRNAs that require either STAT3 or STAT5 to induce apoptosis. In EOC, miRNA-519a promotes apoptosis of SKOV3 cells by directly targeting the 3’UTR of STAT3, which results in a decrease of the mRNA and protein expression levels of STAT3, Mcl-1, and BCL-_XL_ [58] (For a detailed list, see Table A1).

### 3.2. Proliferation

EOC is one of the few cancer types that is partially regulated by hormones. IL-6-induced STAT3 phosphorylation levels were found to be increased in cells treated with follicle-stimulating hormone (FSH), luteinizing hormone (LH), 17β-estradiol or estrogen and it facilitated cell proliferation in human ovarian surface epithelial (HOSE) and ovarian cancer (OVCA) cell lines [59]. In the case of ovarian cancer cell proliferation, Leptin, a hormone secreted by adipose tissue interlinks obesity and EOC progression [31,60]. At the molecular level, Leptin simultaneous increased pY-STAT3 (Tyr705) and nuclear localization of Estrogen Receptor (ER)α, and induced STAT3 binding to ERα that resulted in a significant increase in cell proliferation and migration [32,53,61]. Also, Leptin promoted ovarian cancer proliferation by induced phosphorylation of STAT5 in SKOV3 and A2780 cells, which mediated leptin-induced expression of miR-182 and miR-96 and subsequent inhibition of Forkhead box O3 [62]. Prolactin-mediated pY-STAT5 activation complexes with tumor suppressor *BRCA1* in the nucleus, and this complex hinders the transcription of cell-cycle inhibitor p21, leading to increased proliferation [63]. Treatment of metastatic ovarian carcinoma cell line CaOV-3 with Leptin receptor blockers: SHLA and Lan-2 resulted in a predominantly inhibitory effect on STAT3 phosphorylation and downregulated cell proliferation through blocking *Cyclin D1* and *E2F1* [64].

Expression of phosphorylated STAT3 coupled with Ki-67 expression, was found to be increased in primary human ovarian carcinoma, particularly in patients with high nuclear expression of pY-STAT3 exhibited poor prognosis [15]. EOC cells cultured under hypoxic conditions revealed higher levels of pY-STAT3 (Tyr705); however, with knockdown of STAT3 expression, the proliferation rate of cancer cells was significantly reduced [65]. Alternatively, the hyperbaric oxygenation method (a systemic increase of dissolved oxygen delivery in serum) substantially decreased pY-STAT3 (Tyr705) levels, along with decreased tumor volume in a murine xenograft model [66]. Recently, it was found that activated STAT3 may deploy specific microRNAs to promote ovarian cancer cell proliferation and to generate associated phenotypes. For instance, high STAT3 levels in SKOV3 cells increased the levels of oncogenic microRNA-216a, which in turn directly targeted tumor suppressor phosphatase and tensin homolog (PTEN) expression and rendered cisplatin resistance [67].

Interference of STAT3 activity has been employed as a strategy to hinder ovarian cancer cell proliferation. In vitro and in vivo studies with ovarian cancer cell lines have shown that siRNA and shRNA mediated STAT3 depletion, downregulated the expression of Cyclin D1, Survivin as well as reduced tumor weight [68,69]. Another study has shown that siRNA mediated STAT3 downregulation suppressed SKOV3 cell growth and arrested the cell cycle in the G1 phase [70]. Several studies have highlighted that plant-derived phytochemicals, such as Pterostilbene, Cryptotanshinone and Curcumin, suppressed STAT3 activation, thereby reduced cancer cell proliferation and have been proposed as possible adjuvants of conventional chemotherapy [71,72,73].

### 3.3. Angiogenesis 

In general, tumor outgrowth exceeding 2 mm in diameter must gain access to an increased supply of oxygen and nutrients. These requirements are fulfilled through angiogenesis, a process that involves the formation of new blood vessels from the existing vasculature. Therefore, tumor angiogenesis is a hallmark of cancer that promotes tumor progression and metastasis. Tumor cells cause an angiogenic switch by secreting pro-angiogenic proteins and/or repressing the expression of anti-angiogenic factors. Most notably, vascular endothelial growth factor (VEGF) and its receptors are crucial in instigating angiogenesis in tumor cells, exert vascular permeability activity and stimulate cell migration in macrophage and endothelial cell populations. Activated STAT3 and STAT5 regulate the expression of VEGF and increased angiogenesis in a variety of cancer, including EOC [14,74,75]. Immunohistochemical staining of patient-derived ovarian epithelial carcinoma tissues identified significant overlap of expression between VEGF, pY-STAT3, and pY-STAT5 in ovarian carcinoma cells, compared to the benign and normal group [14]. In ovarian clear cell carcinoma, immunohistochemical analysis of primary tumors showed high nuclear expression of pY-STAT3 and HIF1α [76]. It is known that IL-6 signals via STAT3, not only directly induces the transcription of VEGF, but also activates expression of downstream gene HIF1α, where HIF1α is a paramount transcription factor controlling VEGF expression [77]. These studies indicate an IL-6/STAT3/HIF1α/VEGF autocrine activation loop in EOC, especially clear cell carcinoma histotype. (For a detailed list, see Table A2).

Targeting STAT3 activation in EOC directly or indirectly affected angiogenesis. Extract from cinnamon was a potent inhibitor of VEGF secretion, inhibited the expression and phosphorylation of STAT3 and AKT, suppressed HIF1α expression as well as significantly reduced tumor growth and blood vessel formation in mice models [78]. 3,3’-Diindolylmethane (DIM; an active metabolite found in cruciferous vegetables) treatment blocked IL-6-induced STAT3 phosphorylation, attenuated angiogenesis by suppressing HIF1α and VEGF expression in SKOV3 cells [79]. In the same study, oral administration of DIM in combination with intraperitoneal administration of cisplatin treatment reduced tumor volume by 65% due to downregulation of pY-STAT3, STAT3, and Mcl-1 levels with simultaneously increased cleavage of Caspase 3 and poly ADP-ribose polymerase (PARP) activity. It is important to note that clinical trials of EOC patients treated with the VEGF inhibitor aflibercept revealed more reduced survival rates in patients harboring high levels of circulating IL-6, indicating that IL-6/STAT3 activation in tumor cells may provide a survival gain during anti-VEGF treatment [80]. The fact that STAT3 is critically involved in the VEGF pathway and tumor angiogenesis indicates that blockade of STAT3 is a therapeutic target to heighten an effective antiangiogenic treatment in EOC.

### 3.4. Tumor Progression and Metastasis

Cancer metastasis is a multi-step process that occurs when a selected group of cells detach from the primary tumor, utilize blood or lymphatic system to gain access into circulation, and instigate tumor colonies in secondary organ sites. The metastatic pattern of EOC differs from that of most other epithelial malignant diseases. After the direct extension, EOC most frequently disseminates via the transcoelomic route, forming diffuse multifocal intraperitoneal nodes and malignant ascites [81]. Accumulation of pY-STAT3 expression coupled with loss of protein inhibitor of activated STAT3 (PIAS3) in fallopian tube secretory epithelial cells displayed common peritoneal metastatic nodules that eventually led to the progression of HGSC. *AKT2,* one of the most frequent amplicon alterations in HGSC, activates PKM2–STAT3/NF-κB axis, ultimately results in increased migratory and invasive potential of EOC cells as well as promoting lung metastasis in mouse models [82].

Several hypotheses have been put forward to comprehend the multistep process of metastasis. Yet, recapitulation of an embryonic cell differentiation program known as epithelial to mesenchymal transition (EMT) has gained recent advances to support its inevitable role. Briefly stating, EMT is a process when epithelial cells lose their signature characteristics like cell-cell adhesion to acquire a mesenchymal phenotype with migratory and invasive behavior [83]. Apart from its role in EOC metastasis, EMT can promote tumor initiation, stemness, and chemoresistance in EOC [84]. STAT proteins have been well documented to play a role in fostering EMT in several cancer entities [85,86,87]. In EOC, inhibition of EGFR or STAT3 activity reduced N-cadherin, Vimentin expression, decreased colony-forming ability, cell motility, and migration behavior [35,38]. Also, extracellular heat shock protein (HSP90) promotes the binding of STAT3 to the TWIST1 promoter and thereby increasing TWIST1 transcription, and these effects were diminished after HSP90 inhibitor treatment [88].

Another critical aspect of EMT that strongly associates with STAT proteins is their role in self-renewal. STAT3 and STAT5 have essential roles in regulating cancer stem cells (CSCs) of EOC [89]. Over 90% of ascites cells derived from EOC patients showed activated pY-STAT3 signaling (Tyr705), which increased the migratory potential of EOC cells, and it also increased widespread peritoneal metastasis [90,91]. Tumor spheroids isolated from the ascites of recurrent EOC patients are enriched with tumor cells overexpressing STAT3 compared with cells isolated from the ascites of chemotherapy-naïve patients [92]. EOC spheroids serve as the vehicle for ovarian cancer cell dissemination in the peritoneal cavity and represent a significant impediment in the efficacy of chemotherapy agents [93]. Mechanistically, STAT3 signaling regulates ovarian CSCs by targeting miR-92a/DKK1 and subsequently activating Wnt/β-catenin signaling [18]. Therefore, inhibition of STAT3 signaling effectively eliminates the formation of the metastatic niche, and it suppresses cancer cell persistence after chemotherapy.

Extracellular matrix (ECM) is a complex network of the microenvironment, stabilized through structural proteins such as Laminins, Collagens, and Fibronectins, that holds epithelial-derived cancer cells, endothelial and stromal cells in proximity. Degradation of ECM is a characteristic feature of disseminating cancer cells to determine whether metastatic tumors form or not. Activation of STAT3 in several cancers contributed towards degradation of ECM, primarily mediated through increased activity of matrix-degrading Matrix Metallo-Proteinases (MMPs). In EOC cells, ligand-mediated and stress hormone (norepinephrine) mediated activation of STAT3, via nuclear translocation, induced MMP-2 and MMP-9 expressions and siRNA mediated STAT3 silencing declined MMPs release, denoting the direct regulation of MMPs by STAT3 [30,94,95,96]. ECM component hyaluronan (HA) associated with CD44 to mediate nuclear Nanog-STAT3 interaction, that activated EMT, increased cell migration and invasion and also triggered the expression of MDR1, which rendered multi-drug resistance to EOC cells [97,98]. A BET bromodomain and extra-terminal inhibitor, i-BET151, reduced viability migration and invasion of EOC cells as well as decreased expression of MMP-2 and MMP-9 expression, highlighting the potential usage of this inhibitor as a treatment strategy [99]. siRNA mediated inhibition of IL-6R or anti-human IL-6R antibody (tocilizumab) reduced pY-STAT3 and MMP-9 expression levels suggesting that interference of STAT3 signaling in ovarian clear cell carcinoma could be an effective therapeutic strategy [100]. (For a detailed list, see Table A3, Table A4 and Table A6).

Growing evidence emphasizes the role of nano-sized secretory vesicles known as “exosomes” released from the plasma membrane to facilitate intercellular communication during various physiological processes such as antigen presentation or exchange of membrane proteins. In particular, exosomes are highly implicated in mediating cancer metastasis, angiogenesis, and drug resistance [101,102]. EOC-derived exosomes changed the morphology of human peritoneal mesothelial cells to a mesenchymal phenotype, through CD44 internalization, and blockage of exosome release, which suppressed ovarian cancer invasion [103]. Besides, hypoxia is known to drive excessive exosome release from cancer cells in several tumor types [104]. Under hypoxic conditions, EOC cells showed activated STAT3 levels, increased the release of exosomes to promote proliferation, which occurs through altering proteins of the Rab family [105]. A microfluidic ChIP-based exosomes isolation method confirmed elevated pY-STAT3 levels in exosomes isolated from high-grade serous ovarian cancer cell lines and patients, implying that vesicles secreted from cancer patients have activated STAT3 signaling that could foster cancer metastasis [101].

## 4. Tumor Microenvironment

Emerging evidence reveals the presence of vibrant multicellular interactions between malignant and non-malignant somatic cells, which generate a complex milieu referred to as the “tumor microenvironment” [106,107]. The tumor microenvironment encompasses a multitude of distinct cell types, which primarily includes endothelial cells, infiltrating immune cells (tumor-associated neutrophils, T and B lymphocytes, natural killer cells, tumor-associated macrophages, mast cells) and cancer-associated fibroblasts. These non-malignant somatic cells are often engaged in delivering tumor-promoting factors and are involved in every aspect of tumorigenesis as well as in metastasis. Cellular communications between these cell types are governed by the copious release of cytokines, growth factors, inflammatory and matrix remodeling components from the tumor bulk (Figure 1). Although the immune cells of the tumor microenvironment are efficient enough to combat and evade the cancer cells, these cells are instead confined and manipulated by cancer cells to promote their growth and distribution, ultimately influencing the patient’s clinical outcome [108]. Henceforth, understanding the biology of the tumor-host hostile environment becomes inevitable for improving treatment strategies. Several lines of evidence have highlighted the role of STAT3/5 activation in non-malignant somatic cells of the EOC tumor microenvironment and the crosstalk between tumor and host stromal cells.

### 4.1. Non-Immune Stromal Cells

In the tumor microenvironment, non-immune stromal cells comprise endothelial cells, pericytes, fibroblasts, and mesenchymal stem cells. STAT3 pathway facilitates crosstalk between tumor cells and endothelial cells that mediates pro-angiogenic signaling. Conditioned media obtained from EOC cells stimulated rapid, transient STAT3 phosphorylation and nuclear translocation in cord blood CD34^+^ progenitor cells, as well as initiated early capillary-like structure formation in human microvascular endothelial cells [109,110]. CD133^+^ ovarian cancer stem cells cultured on a cell culture matrix formed fluid-conducting tube networks activated NF-κB and STAT3 signal pathways, through autocrine chemokine (C-C motif) ligand 5 (CCL5) upregulation promoting differentiation into endothelial cells [111]. Implantation of HeyA8 and SKOV3 cells into the peritoneal cavity of female nude mice secreted significant levels of IL-6 and soluble IL-6 receptor (sIL-6R), activated STAT3 signaling and facilitated migration of endothelial cells [112].

Cancer-associated fibroblasts (CAFs) are a heterogeneous cell population that is usually identified by their high expression of mesenchymal markers including Vimentin, fibroblast-secreted protein-1 (FSP-1), α-smooth muscle actin (αSMA), and fibroblast activation protein (FAP). CAFs favor tumor growth through increased secretion of cytokines, metabolites, and ECM molecules, thus promoting angiogenesis and they interfere with antitumor immune response [113]. CAFs mediate EOC cell proliferation through NF-κB and JNK signaling activation. Moreover, they release VEGF to promote tumor angiogenesis [114]. CAFs induce EMT in EOC cell lines through IL-6/JAK2/STAT3 pathway, which subsequently renders resistance to paclitaxel treatment [115]. Also, CAFs release chemokine CCL5 to increase STAT3 and Akt phosphorylation that mediate cisplatin resistance in EOC cells [116,117].

In the EOC tumor microenvironment, the intra-abdominal fat deposition is regarded as a significant source of cytokines and has been shown to stimulate growth and promote EOC metastasis [118]. Conditioned media obtained from subcutaneous and visceral fat-derived adipose stromal cells enhanced growth and migration of EOC cells, through activation of JAK2/STAT3 signaling pathway [119]. Studies demonstrating the distinct regulation of STAT3 in other non-immune stromal cell populations are quite preliminary and need further in-depth investigations.

### 4.2. Immune Function

T lymphocytes play a crucial role in stimulating the adaptive immune response to target the expanding tumor mass. Depending on the nature of the tissue, T-cells can either be pro- or anti-tumorigenic. Tumor-infiltrating T-cells exist as distinct populations, and here we highlight a few main types of T-cell players that are regulated through STAT3/5 signaling.

CD4^+^ Th17 cell population in EOC is identified to be pro-tumorigenic. EOC cells secrete cytokines, including IL-1β, IL-6, and IL-23, which are involved in the expansion of CD4^+^ Th17 cell population (Figure 1), through activation of STAT3 [120,121]. Phosphorylation and acetylation profiles of STAT3 determine the differentiation and polarization of CD4^+^ Th17 cells that form the majority of tumor infiltrated T cells [122,123]. An increase in the Th17 cell population sustains the secretion of more cytokines (IL-17, IL-23) that eventually stimulates the release of angiogenesis factors VEGF and TGFβ in fibroblasts and endothelial cells. One study indicated exosomes transfer miRNAs, including miR-29a-3p and miR-21-5p, to synergistically induce the Treg/Th17 cell imbalance through direct targeting of STAT3 in CD4^+^ T cells [124]. Thus, active STAT3 in CD4^+^ T cells generates an inflammatory environment around the budding tumor aids its growth by stimulating angiogenesis and abrogates antitumor response. (For a detailed list, see Table A5).

Another subpopulation of CD4^+^ cells known as regulatory T (Treg) cells that are involved in the dampening of antitumor activity in the tumor microenvironment. Accumulation of Treg cells in tumors and ascites of patients with EOC showed reduced survival rate [125]. STAT3 and STAT5 are known to bind to the promoter and increase the transcription of FOXP3, which is essential for the conversion of naive CD4^+^ T cells into CD4^+^CD25^+^ FOXP3^+^ Treg cells [126,127]. In response to IL-6 and IL-12 stimulation, STAT3 also positively regulates immune check point proteins such as programmed cell death receptor 1 (PD-1), programmed death-ligand (PD-L1) expression through direct promoter binding in CD4^+^ T cells. PD-1/PD-L1 axis promotes the differentiation of CD4^+^ T cells into FOXP3^+^ Treg cells (Figure 1), and EOC patients with high PD-1 expression showed poor prognosis [128]. Also, STAT3 induces the expression of IL-10 and TGF-β1, through direct promoter binding, which ultimately restrains the tumor-suppressive role of CD8^+^ effector T-cell function and dendritic cells [129,130]. Another important finding revealed that STAT3 sequesters STAT1 through cytoplasmic heterodimerization and hinders STAT1 mediated transcription of major histocompatibility complex (MHC) class I genes [131]. Reduced surface expression of MHC class-I genes in cancer cells pose unfavorable presentation of cancer cells to CD8^+^ effector cells and thereby diminishing tumor immunosurveillance.

CD8^+^ T lymphocytes in the tumor microenvironment play a crucial role in combating cancer cells through antigen-specific blockage of immunosuppressive Treg cells. This blockage is regulated by STAT3 through the secretion of IFN-γ [132]. Using the Mx1-Cre-loxP mice model, Kortylewski et al. showed that STAT3^−/−^ CD8^+^ T cells can secrete increased IFN-γ levels, which displayed markedly enhanced tumor-suppressive functions of dendritic cells, natural killer cells and neutrophils [133]. Moreover, a study reported that when tumor supernatants (from EOC cell lines OVCAR3, CAOV3 and SKOV3) were co-cultured with CD8^+^ T cells reduced STAT5 phosphorylation which diminished CD8^+^ T cell proliferation [134]. EOC patients with increased levels of CD8^+^ T cells infiltration and a high CD8^+^/Treg ratio showed favorable prognosis [135,136].

### 4.3. Tumor-Associated Macrophages

Tumor-associated macrophages (TAMs) form a prominent component of the inflammatory tumor microenvironment, which primarily performs a pro-tumorigenic role, including the release of proangiogenic cytokines, matrix proteases and growth factors, suppression of adaptive immunity, self-renewal and chemotherapeutic resistance of cancer stem cells [137]. siRNA-mediated STAT3 inhibition in macrophages increased IL-6 and IL-10 secretion, induced Cyclin-D1 mediated cell proliferation in co-cultured SKOV3 cells [138]. Other studies have shown that the interaction of TAMs and ovarian cancer stem-like cells (OCSLCs) being involved in the occurrence, recurrence, and multidrug resistance of ovarian cancers [139]. The OCSLCs co-cultured with macrophages also induces SKOV3 cell stemness via IL-8/STAT3 signaling [140].

Macrophages exist as a heterogeneous population, that can be broadly classified into two main phenotypes: classically activated M1 macrophages and alternatively activated M2 macrophages. The polarization of M1 or M2 macrophages is highly dependent on the cytokines involved in their activation. M1 macrophages are activated by cytokines such as interferon-γ (IFNγ), IL-12 and they play a crucial role in recruiting T cells to the tumor microenvironment. M1 macrophages enhanced immune response in order to restrain tumor development. On the contrary, M2 macrophages are activated by Th2 cytokines (IL-4, IL-10, and IL-13). M2 macrophages interfered with the antitumor activity of T cells or NK cells. EOC patients with a high number of M2 CD68^+^ and CD163^+^ macrophages were associated with advanced stage and poor progression-free and overall survival [141]. Accordingly, ovarian cancer patients with high M1 to M2 ratio of TAMs were associated with extended survival rate, indicating the pro-tumorigenic role of M2 macrophages [142]. Moreover, ascites from EOC patients polarized macrophages toward the M2 phenotype through STAT3 activation, while non-EOC did not activate STAT3. Also, the expression of transmembrane protein B4-T7 in TAMs, but not in ovarian carcinoma cells suppressed the T-cell immune response, inversely correlated with patient survival [143]. These studies have demonstrated that STAT3 mediated polarization of macrophages is crucial in determining the pro-tumorigenic nature of tumor microenvironment. (For a detailed list, see Table A5).

## 5. Targeting of the STAT3/STAT5 Signaling Pathway in EOC

In principle, targeting the constitutive activation of STAT3/5 could be approached in various ways. These strategies include (1) inhibiting the upstream of STAT3 activation pathway (for example, ligands antagonist, JAK or SRC inhibitor) (2) blocking the SH2 domain that inhibit STAT dimerization (3) inhibiting the translocation of phosphorylated dimerized STAT3 into the nucleus (4) inhibit the binding of activated dimerized STAT to DNA [144,145]. Table 1 lists the names and properties of agents targeting JAK/STAT that are FDA approved or in clinical development.

The anti-human-IL-6 antibody Siltuximab substantially reduced nuclear pY-STAT3 expression in IL-6-producing intraperitoneal EOC xenografts [146]. Siltuximab significantly inhibited the tumor growth of IGROV1 intraperitoneal xenograft, accompanied by reductions in angiogenesis and macrophage infiltration. In phase 2 clinical trial, single-agent Siltuximab showed modest response rate (5.6%) with reductions in serum CCL2, CXCL12, and VEGF, in recurrent, platinum-resistant diseases [146]. In phase 1 dose-finding trial, the anti-IL-6R monoclonal antibody Tocilizumab combined with carboplatin/pegylated liposomal doxorubicin and interferon α2b in patients with advanced EOC was studied. A functional blockade of IL-6 signaling by Tocilizumab decreased pY-STAT3 by both myeloid cells and the different populations of T cells, leading to increased production of the tumor-immunity promoting cytokine secretion of IL-12 and IL-1β [147]. The clinical benefit was observed in 85% (17/20) in a patient with recurrent EOC. The revamping of immunity support for better tumor control by targeting IL-6/STAT3 signaling in EOC.

JAK inhibitors have been developed in recent years. The utilization of JAK inhibitors has been attempted to interfere with JAK-mediated STAT3/5 activation and evaluate therapeutic efficacy in various EOC models. Ruxolitinib, a potent oral JAK1, and JAK2 inhibitor has been approved by FDA to treat myelofibrosis in 2011. Ruxolitinib significantly inhibited pY-STAT3 in EOC cells. Single-agent Ruxolitinib suppresses EOC tumor growth in mice. More importantly, Ruxolitinib substantially enhances the anti-tumor activity of chemotherapy agents, including paclitaxel, cisplatin, carboplatin, doxorubicin, and topotecan in EOC cells. In an OVCAR-8 murine model, Ruxolitinib synergistically increased tumor control by paclitaxel [147]. A phase I/II clinical trial of EOC has been conducted with combination with or without paclitaxel and carboplatin since 2016 and is under recruiting now (ClinicalTrials.gov Identifier: NCT02713386). The FDA also approved other JAK inhibitors, tofacitinib (pan JAK with preferentially selectively JAK 3/1) in 2016 and baricitinib (selectively JAK 1/2) in 2018 for the treatment of rheumatoid arthritis [148]. However, there are, as yet, no studies or trials mentioning the relationship between these two and gynecologic cancers. Itacitinib, another JAK 1/2 inhibitor, has been tested in non-small cell lung cancer, lymphoma, and pancreatic cancer. A previous terminated trial showed an acceptable safety profile in combination with nab-paclitaxel and gemcitabine [149]. It has been tested in multiple cancers including endometrial cancer and breast cancer in the proceeding phase Ib/II clinical trial (ClinicalTrials.gov Identifier: NCT02646748). In the preclinical stage, paclitaxel and Momelotinib (ATP-competitive inhibitor of JAK1/2, previously named CYT387) inhibited JAK2/STAT3 activation, reduced tumor burden, abrogated cancer stem cell expressions and prolonged disease-free survival in a murine xenograft model [89,150]. MLS-2384 is a synthetic 6-bromoindirubin derivative with potent dual JAK/SRC inhibitory activity in EOC cells. In vitro, MLS-2384 suppresses the viability of A2780 cells, which is consistent with the inhibition of phosphorylation of JAK2, SRC, and STAT3 [151]. Although these JAK inhibitors are widely used in medicine, the side effect of JAK inhibitors is significant, ranging from immunosuppression to organ toxicity. AZD1480, a selective JAK2 inhibitor, reduced tumor growth, decreased peritoneal dissemination and diminished ascites production in a murine model for advanced EOC [152]. However, the previous phase I study of the solid tumor was terminated due to neuropsychiatric side effects [153].

Pilot studies focusing on analogs from natural compounds have been developed to inhibit STAT3 activation. The anticancer analogs from curcumin, diarylidenyl piperidone (DAP) derivatives such as HO-3867, HO-4200, HO-4318, inhibit STAT3 activity and sensitize drug-resistant ovarian carcinoma and clear cell carcinoma cells to paclitaxel or cisplatin [52,156,157,158]. Among them, HO-3867 is the most studied and effective in selectively targeting STAT3 and inhibiting EOC tumor growth. Specifically, HO-3867 effectively blocked ascites-mediated activation of STAT3 in EOC cells, inhibited invasion, and metastasis in a murine orthotopic EOC model [91]. Additionally, HO-3867 targets hypoxia-stimulated pY-STAT3 (Tyr705) via the ubiquitin–proteasome degradation pathway, leading to tumor growth suppression in xenograft mice and the downregulation of proteins involved in cell survival, proliferation, and angiogenesis [159]. Furthermore, HO-4200 and HO-4318 significantly inhibited fatty acid synthase and pY-STAT3 and decreased the expression of STAT3 target proteins in primary platinum-resistant EOC cells, resulting in decreased expression of Ki67 and VEGF in ex-vivo human tumor specimens [156].

Direct inhibition of STAT3 activity through the interference of SH2 domain dimerization, nucleus transportation, or DNA binding is another highly investigated therapeutic strategy. However, most of the molecules proposed are still in preliminary stage of development for EOC treatment. Small-molecules such as decoy oligo-deoxy-nucleotides (ODN) inhibit STAT3 activation by blocking the pY-STAT3 nuclear translocation for subsequent transcriptional target gene activation. STAT3-ODN has been examined in several in vitro EOC cell models and demonstrated significant inhibition of invasiveness and enhancement sensitivity to chemotherapeutic agents [160,161,162].

Lastly, proteomic analysis has identified RELA and STAT5 as two vital proteins associated with carboplatin resistance in HGCS patients. Small-molecule mediated inhibition of NF-kB (by BMS-345541) and STAT5 (by dasatinib) synergistically sensitized carboplatin-resistant EOC cells towards carboplatin treatment [19].

## 6. Conclusions

Research in past decades has facilitated our comprehension of the critical roles of aberrant STAT3/STAT5 activation in EOC cells as well as in the tumor microenvironment. The STAT3/STAT5 signaling dysregulates a plethora of cellular processes in EOC, which results in uncontrolled cancer cell proliferation, induction of angiogenesis, promotion of metastasis factors, and suppression of host immune response. In addition to extracellular signals that activate STAT, phosphorylated STATs positively regulate the expression of interleukins and growth factors, generating a vicious autocrine feedback loop that sustains the constitutive STAT3/STAT5 signaling cascade. Accordingly, multiple studies have provided ample evidence to show that interfering with STAT3/STAT5 signaling, through knockdown or inhibitor intervention, resulted in antitumor effects in both in vitro and in vivo animal models carrying human tumors. However, none of the existing candidate compounds showed anti-tumor efficacy in EOC patients. Given the importance of STAT3/STAT5 as a promising target, future research should explore inhibitors against upstream regulators for the development of clinically useful anticancer therapeutics. A histotype-specific approach to target ovarian clear cell carcinoma with the STAT3 pathway might be an avenue worth pursuing.

Other abbreviations: SHC; SRC homology and collagen family, GRB2; Growth factor receptor-bound protein 2, SOS; Son of sevenless, NK; Natural killer, PIAS; Protein inhibitor of activated STAT, PD-1; Programmed cell death protein 1, PD-L1; Programmed death-ligand 1, PTPN11; Tyrosine-protein phosphatase non-receptor type 11, PTPRT; Protein Tyrosine phosphatase receptor type T, MHC-I; Major histocompatibility complex class I, VEGF; Vascular endothelial growth factor, RTK; Receptor Tyrosine kinase.

## Figures and Tables

**Figure 1 cancers-12-00024-f001:**
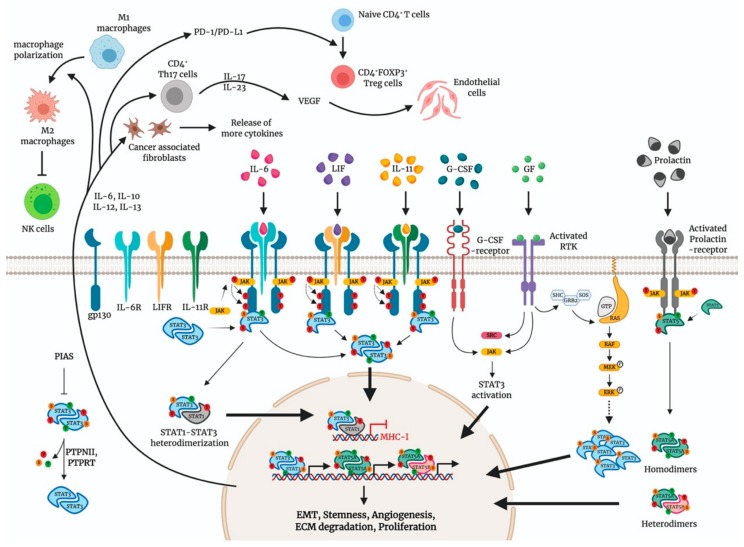
Signal transducers and activators of transcription (STAT)3 and STAT5 signaling in epithelial ovarian cancer (EOC) and tumor microenvironment. Distinct families of cytokines such as Interleukins (IL-6,IL-11) and leukemia inhibitory factor (LIF) bind to their homodimeric cognate receptors IL-6R, IL-11R and LIFR respectively, and share a signal-transducing receptor gp130. Janus kinase (JAK) phosphorylate gp130 to enable docking and phosphorylation of STAT3 at Tyrosine (symbol Y or Tyr) residue 705. Tyrosine phosphorylation of STAT3 can also be mediated by activation of other oncogenic proteins including growth factor (GF)-mediated receptor Tyrosine kinase (RTK) activation, granulocyte colony-stimulating factor (G-CSF)-mediated activation, SRC and RAS/MEK/ERK pathway. Phosphorylated STAT3 dynamically undergo dimerization and nuclear translocation to trigger STAT3-mediated transcription of target genes. Binding of Prolactin to its receptor facilitate JAK-mediated phosphorylation of STAT5A and STAT5B at Tyr residue 694 and Tyr residue 699, respectively, leading to homodimerization or heterodimerization before nuclear translocation for target gene activation. STAT3 and STAT5 signaling in cancer cells release more cytokines into tumor microenvironment that generate a plethora of immune-compromising functions (highlighted in the main text). Figure created with Biorender.com.

**Table 1 cancers-12-00024-t001:** Agents targeting the Janus kinase/signal transducers and activators of transcription (JAK/STAT) that are U.S. Food and Drug Administration (FDA) approved or in clinical development.

Category	Drug	Target	Phase	Histology	Remark	References
**Upstream ligand**	Siltuximab (CNTO 328)	IL-6	Phase II	19 serous, 1 CC	Modest RR as monotherapy Reduce angiogenesis and macrophage infiltration Clinial approval for Castleman’s disease.	[146]
	Tociclizumab	IL-6R	Phase I	15 serous, 3 CC, 3 EM	A comination with interferon-alpha showed clinical benefit and immune cell change, but not powered. Proceed to phase II in EOC. Clinical approval for RA, juvanile idiopathic arthritis, CAR T-cell-induced cytokine-release syndrome	[154]
**Upstream kinase inhibitors**	Ruxolitinib (INC424)	JAK 1, JAK 2	Phase I/II	Not specified	In EOC, synergistically increase tumor control with multiple chemotherapy agents (see context) Phase I/II study, combined with paclitaxel and carboplatin in advanced EOC Clinical approval for MF, PCV	[147] NCT02713386.
	Tofacitinib (CP690550)	JAK3>JAK1>>JAK2	Not tested in EOC	-	Clinical approval for RA	
	Baricitinib (INCB28050)	JAK 1, JAK 2	Not tested in EOC	-	Clinical approval for RA	
	Itacitinib (ABT494)	JAK 1, JAK 2	Not tested in EOC	-	Phase Ib/II study, combined with pembrolizumab to test multiple cancers including endometrial cancer and breast cancer EMA orphan approval for GVHD.	NCT02646748
	Momelotinib (CYT387)	JAK 1, JAK 2	Preclinical	Serous and CC cell lines	Reduce tumor burden in combination with paclitaxel compared with paclitaxel only arm in xenograft mice. No significant effect was noted on CYT378 only treatment compared with control. Phase III trials and FDA Fast-track designation in MF patients	[150,155]
	MLS2384	c-SRC, JAK 2, TYK2	Preclinical	Not specified	Suppress cell viability in diverse human cancer cell lines in vitro.	[151]

Abbreviations: AE; adverse effect, CAR; chimeric antigen receptor, CC; clear cell, DFS; disease-free survival, EM; endometrioid, EMA; European medicines agency, GVHD; graft-versus-host disease, IP; intraperitoneal, MF; myelofibrosis, PCV; polycythemia vera; RR; response rate; TYK 2; tyrosine kinase 2.EOC: epithelial ovarian cancer, RA; rheumatic arthritis, IL; interleukin.

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
