# Peer review of "Activation of STAT3 and STAT5 Signaling in Epithelial Ovarian Cancer Progression: Mechanism and Therapeutic Opportunity"

_cancers, 2019, doi:10.3390/cancers12010024_

Round 1

Reviewer 1 Report

Wu et al. provide a comprehensive review of the literature on the role of STAT3 and STAT5 in epithelial ovarian cancer (EOC). The manuscript is well structured and covers all relevant aspects from activation of STAT3/5, their function in EOC, the role of the tumor microenvironment including immune cells and finally targeting of STAT3/5 including an overview on clinical trials. However, while referring to a multitude of published work, a more critical evaluation of individual contributions is somewhat missing.
The figure is appealing and nicely summarizes the main aspects of the review.

Minor points:

Page 2:
“(For a comprehensive review, please visit doi:10.1038/nrc2734)”
Should be cited the same way as the other references.
“... associated with both solid and hematopoietic cancers [8-10].”
Refs. 8-10 do not cover hematopoietic cancers

Page 3:
“STAT5 Tyr residue 694 and 699” should read:
STAT5: Tyr694 in STAT5A, Tyr699 in STAT5B
“... mutant p53 in EOC cells [28].”
Ref. 28 refers to prostate cancer.

Page 14:
“STAT5 (dasatinib)”
Dasatinib is a Bcr-Abl inhibitor used for the treatment of CML. How is dasatinib related to EOC?

The English language is generally acceptable, but there a many errors in grammar and style throughout the text that require careful proof-reading.

A few examples:
Page 3:
“The IL-6 family of cytokines is one of the major immunoregulatory cytokines.” should read:
The IL-6 family of cytokines is one of the major families of immunoregulatory cytokines
“Vascular endothelial growth factor (VEGF), a key mediator of angiogenesis in EOC, induces p-STAT3 through the binding VEGF receptor 2 (VEGFR2) in EOC cells.” should read: through the binding of VEGF receptor 2 or through binding to VEGF receptor 2

Page 6:
“The fact that STAT3 critically involved in ...” should read:
The fact that STAT3 is critically involved in ...

Page 10
“... inhibiting the dimerized STAT3 into the nucleus” should read:
... inhibiting the translocation of dimerized STAT3 into the nucleus

Page 13:
“JAK inhibitors has been developed in recent years.” should read:
JAK inhibitors have been developed in recent years.

There are many mistakes in the references section that need to be corrected. Ref. 80 may serve as an example.

Author Response

Author's Notes to Reviewer 1

Page 2: “(For a comprehensive review, please visit doi:10.1038/nrc2734)” Should be cited the same way as the other references. “... associated with both solid and hematopoietic cancers [8-10].” Refs. 8-10 do not cover hematopoietic cancers

-> We agree with the reviewer’s comment and revised accordingly.

Page 3: “STAT5 Tyr residue 694 and 699” should read: STAT5: Tyr694 in STAT5A, Tyr699 in STAT5B. “... mutant p53 in EOC cells [28].” Ref. 28 refers to prostate cancer.

-> We agree with the reviewer’s comment and revised accordingly.

Page 14: “STAT5 (dasatinib)” Dasatinib is a Bcr-Abl inhibitor used for the treatment of CML. How is dasatinib related to EOC?

-> Dasatinib is an FDA-approved inhibitor used for Bcr-Abl, and also an inhibitor of Src family kinases that can be used to suppress the STAT5 pathway.

The English language is generally acceptable, but there are many errors in grammar and style throughout the text that require careful proof-reading.

A few examples:

Page 3: “The IL-6 family of cytokines is one of the major immunoregulatory cytokines.” should read: The IL-6 family of cytokines is one of the major families of immunoregulatory cytokines.  “Vascular endothelial growth factor (VEGF), a key mediator of angiogenesis in EOC, induces p-STAT3 through the binding VEGF receptor 2 (VEGFR2) in EOC cells.” should read: through the binding of VEGF receptor 2 or through binding to VEGF receptor 2.

Page 6: “The fact that STAT3 critically involved in ...” should read: The fact that STAT3 is critically involved in ...

Page 10: “... inhibiting the dimerized STAT3 into the nucleus” should read: ... inhibiting the translocation of dimerized STAT3 into the nucleus

Page 13: “JAK inhibitors has been developed in recent years.” should read: JAK inhibitors have been developed in recent years.

-> We have reviewed and revised accordingly.

There are many mistakes in the references section that need to be corrected. Ref. 80 may serve as an example.

-> We have reviewed and revised accordingly.

Reviewer 2 Report

Overall, this review is quite well-written in details and in depth, however there are major inconsistencies and major gaps which should be adressed by the authors should before this can be published.

The authors have put a lot of efforts in describing the role of STAT3 in EOC, whereas the crucial role/effect of STAT5 in EOC is rather weak. E.g. “activation of STAT5”, "function of STAT5" in proliferation, angiogenesis, tumour progression and metastasis section, "tumour microenvironment" and "targeting STAT5" etc. STAT5 should be emphasized more in the above-mentioned sections as the title includes both STAT3 “and STAT5” signalling in EOC.

The authors should furthermore describe the known or possible synergistic or antagonistic roles of STAT3 and STAT5 in cancer and especially in EOC in a separate paragraph.

The authors should do a graphical abstract depicting the main functions of STAT3/STAT5 in EOC

Unphosphorylated STATs also form dimers. To be more cautious and precise, the authors had better rephrase the sentence mentioning the dimerization of phosphorylated STATs in the introduction part.

The whole article mainly addresses on pSTAT3 Y705. Are there any other studies regarding the role or effect of pSTAT3 S727 in EOC, except that it may contribute to platinum-resistance in some EOC cell lines? Discuss the possible role of pSTAT3 S727, which is really important elsewhere.

The authors should keep it consistent when mentioning certain proteins, genes and phosphorylation site. E.g. IL6 or IL-6, pSTAT3 or p-STAT3, BCL2 or BCL-2, Tyr705 or Tyr-705 etc.? The authors should go carefully through the article, unify them and avoid such sloppy mistakes.

The authors should be aware that when mentioning genes/proteins in human or other species, for instance genes in humans, should be in italic and capital. Please change them throughout the manuscript accordingly.

In 2.6, the abbreviation of Tyr is Y and hence, it should be pY372 instead of pT372.

BET inhibitors have been shown to reduce STATs-super-enhancer-associated transcription in some neoplasms, making them novel promising targets in therapy. Are there any studies describing the same phenomenon about STAT3/5 super-enhancers in EOC?

The authors should do a graphical summary to depict the role of STAT3/5 signalling in EOR in the context of the Hallmarks of Cancer (Hanahan and Weinberg 2011) as well as the possible respective treatment strategies for EOC in the present an possibly in the future.

Author Response

Author's Notes to Reviewer 2

The authors have put a lot of efforts in describing the role of STAT3 in EOC, whereas the crucial role/effect of STAT5 in EOC is rather weak. E.g. “activation of STAT5”, "function of STAT5" in proliferation, angiogenesis, tumour progression and metastasis section, "tumour microenvironment" and "targeting STAT5" etc. STAT5 should be emphasized more in the above-mentioned sections as the title includes both STAT3 “and STAT5” signalling in EOC.

-> In our literature review, there wasn’t sufficient information being published on delineating STAT5 function in EOC progression. However, we have tried our best to include the past studies elucidating the role of STAT5 in EOC. Based on the reviewer’s comment, we have included few more studies (which can be found in Track Changes) for your reference.

The authors should furthermore describe the known or possible synergistic or antagonistic roles of STAT3 and STAT5 in cancer and especially in EOC in a separate paragraph.

-> Although constitutive activation of STAT3 and STAT5 is observed in several cancer entities, few studies have elucidated the presence of a reciprocal repressive signaling between STAT3 and STAT5. In two breast cancer cell lines, activated STAT3 increased the expression of an oncogenic transcriptional repressor BCL6, whereas overexpression of STAT5 repressed BCL6 expression below basal level (Walker et al., 2009, 2013). Another study reported that STAT3 decreased the release of IL-2, which is the major STAT5-activating cytokine crucial for the Th9 differentiation (Olson et al., 2016). These studies highlight the presence of cross-regulation between STAT3 and STAT5, however such opposing regulation is not documented during EOC progression and therefore require future studies. Henceforth, we have not discussed the cross-regulation between STAT3 and STAT5 in our manuscript.

3.The authors should do a graphical abstract depicting the main functions of STAT3/STAT5 in EOC

-> We have included a new figure – “Figure 2 – Activated STAT3/STAT5 signaling in EOC modulates several hallmarks of cancer” in the revised version to illustrate the main functions of STAT3/STAT5 in EOC.

4.Unphosphorylated STATs also form dimers. To be more cautious and precise, the authors had better rephrase the sentence mentioning the dimerization of phosphorylated STATs in the introduction part.

-> We have revised the sentences to “phosphorylated STAT dimer undergoes conformation change…”.

5.The whole article mainly addresses on pSTAT3 Y705. Are there any other studies regarding the role or effect of pSTAT3 S727 in EOC, except that it may contribute to platinum-resistance in some EOC cell lines? Discuss the possible role of pSTAT3 S727, which is really important elsewhere.

-> We agree with the reviewer that p-STAT3 S727 activation may play an important role in certain cancer types, including glioma and breast cancer. We have made a thorough literature reviews on STAT3 and EOC, and information regarding p-STAT3 S727 activation in EOC was extremely limited.

The authors should keep it consistent when mentioning certain proteins, genes and phosphorylation site. E.g. IL6 or IL-6, pSTAT3 or p-STAT3, BCL2 or BCL-2, Tyr705 or Tyr-705 etc.? The authors should go carefully through the article, unify them and avoid such sloppy mistakes.

-> We have revised accordingly.

The authors should be aware that when mentioning genes/proteins in human or other species, for instance genes in humans, should be in italic and capital. Please change them throughout the manuscript accordingly.

-> We have reviewed and revised accordingly. 

In 2.6, the abbreviation of Tyr is Y and hence, it should be pY372 instead of pT372.

-> We have revised accordingly.

BET inhibitors have been shown to reduce STATs-super-enhancer-associated transcription in some neoplasms, making them novel promising targets in therapy. Are there any studies describing the same phenomenon about STAT3/5 super-enhancers in EOC?

-> Although Hnisz et al found STAT3 as a super enhancer, no direct relationship between BET inhibitor and STAT3 has been revealed. In addition, the JQ-1 BET inhibitor was used to treat ovarian cancer cell lines A2780R and elevated level of STAT3 phosphorylation was observed. Ovarian cancer expressed resistance to BET inhibitor possibly by kinome-mediated process (Hnisz et al., 2013; Kurimchak et al., 2016). It definitely requires more study to prove this finding.

The authors should do a graphical summary to depict the role of STAT3/5 signaling in EOR in the context of the Hallmarks of Cancer (Hanahan and Weinberg 2011) as well as the possible respective treatment strategies for EOC in the present a possibly in the future.

-> The newly generated Figure 2 incorporates the main functions of STAT3/STAT5 in the context of the Hallmarks of Cancer. Regarding the treatment strategies in EOC, in section 5 we have clearly explained the targeting of STAT3/STAT5 in EOC. We feel that to explain about treatment strategies in EOC irrespective of STAT3/STAT5 targeting would be out of the manuscript’s scope and therefore have not included.  

We would like to express our appreciation for the critiques from the reviewers.

Reference

Hnisz, D., Abraham, B.J., Lee, T., Lau, A., Saint-André, V., Sigova, A.A., Hoke, H.A., and Young, R.A. (2013). Super-Enhancers in the Control of Cell Identity and Disease. Cell 155, 934–947.

Kurimchak, A.M., Shelton, C., Duncan, K.E., Johnson, K.J., Brown, J., O’Brien, S., Gabbasov, R., Fink, L.S., Li, Y., Lounsbury, N., et al. (2016). Resistance to BET Bromodomain Inhibitors Is Mediated by Kinome Reprogramming in Ovarian Cancer. Cell Reports 16, 1273–1286.

Olson, M.R., Verdan, F., Hufford, M.M., Dent, A.L., and Kaplan, M.H. (2016). STAT3 Impairs STAT5 Activation in the Development of IL-9–Secreting T Cells. J Immunol 196, 3297–3304.

Walker, S.R., Nelson, E.A., Zou, L., Chaudhury, M., Signoretti, S., Richardson, A., and Frank, D.A. (2009). Reciprocal Effects of STAT5 and STAT3 in Breast Cancer. Mol Cancer Res 7, 966–976.

Walker, S.R., Nelson, E.A., Yeh, J.E., Pinello, L., Yuan, G.-C., and Frank, D.A. (2013). STAT5 Outcompetes STAT3 To Regulate the Expression of the Oncogenic Transcriptional Modulator BCL6. Mol Cell Biol 33, 2879–2890.

Round 2

Reviewer 2 Report

as far as I can see, the revision was performed according to the comments

Author Response

The English was revised by native speaker.